# Prevalence and Molecular Characterization of Bovine Parainfluenza Virus Type 3 in Cattle Herds in China

**DOI:** 10.3390/ani13050793

**Published:** 2023-02-22

**Authors:** Yunxin Ren, Cheng Tang, Hua Yue

**Affiliations:** College of Animal and Veterinary Sciences, Southwest Minzu University, Chengdu 610041, China

**Keywords:** bovine parainfluenza virus type 3, China, prevalence, molecular characterization, phylogenetic analysis, evolution

## Abstract

**Simple Summary:**

BPIV3 is a common pathogen causing respiratory disease in cattle and a significant contributing factor to BRDC; however, data on the prevalence and molecular features of BPIV3 are still scarce in China. The current study sought to investigate the prevalence and molecular features of BPIV3 in BRDC-affected beef cattle and yaks in China, and the findings suggest that BPIV3 genotype C strains, the dominant strains in China, have a broad geographical distribution. Phylogenetic analysis of the HN and genomic sequences reveals Chinese strains with some unique genetic characteristics, which enhances our knowledge of the prevalence and genetic variation in BPIV3.

**Abstract:**

Bovine parainfluenza virus type 3 (BPIV3) is a common respiratory pathogen that causes respiratory illness in cattle and makes a major contribution to the bovine respiratory disease complex (BRDC); however, data on the prevalence and molecular features of BPIV3 are still scarce in China. To investigate the epidemiological characteristics of BPIV3 in China, between September 2020 and June 2022, 776 respiratory samples were received from 58 BRDC-affected farms located in 16 provinces and one municipality. Those were screened for BPIV3 using a reverse transcription insulated isothermal PCR (RT-iiPCR) assay. Meanwhile, the HN gene and complete genome sequence of strains from different provinces were amplified, sequenced, and analyzed. The tests showed that 18.17% (141/776) of samples tested were positive for BPIV3, which originated from 21 farms in 6 provinces. Moreover, 22 complete HN gene sequences and 9 nearly complete genome sequences were obtained from the positive samples. Phylogenetic analysis based on the HN gene and complete genome sequences revealed that the sequences were clustered in one large clade for all Chinese BPIV3 genotype C strains, while overseas strain sequences of BPIV3 genotype C clustered into other clades. Moving beyond the known complete genome sequences of BPIV3 in GenBank, a total of five unique amino acid mutations were found in N protein, F protein, and HN protein in Chinese BPIV3 genotype C strains. Taken together, this study reveals that BPIV3 genotype C strains, the dominant strains in China, have a broad geographical distribution and some unique genetic characteristics. These findings contribute to our understanding of the epidemiological characteristics and genetic evolution of BPIV3 in China.

## 1. Introduction

Bovine parainfluenza virus type 3 (BPIV3) is a respiratory tract pathogen of the genus *Respirovirus* belonging to the family *Paramyxoviridae* [1]. Infection of cattle with BPIV3 causes tissue damage, immunosuppression, and heightened susceptibility to secondary bacterial infections, leading to more severe clinical signs. The resulting disease was described as bovine respiratory disease complex (BRDC), which causes significant economic losses to producers each year [2,3,4]. 

The BPIV3 genome is a negative-strand RNA of 15434–15504 bases in length, encoding six structural proteins (N, P, L, M, F, and HN) and three non-structural proteins (C, D, and V) [5,6]. According to HN gene sequence or complete genome sequence phylogenetic analysis, BPIV3 can be divided into three genotypes, A, B, and C, and genotype C was first identified in China [4]. Genotype C has since been found in Japan, the US, Korea, and Turkey [7,8,9,10]. So far, three BPIV3 genotypes have been found in China [11]. A seroepidemiological study of BPIV3 showed that the antibody positive rates of BPIV3 in the Jilin, Xinjiang, Inner Mongolia, and Shanxi regions were more than 80%, indicating that BPIV3 was widespread in northern China. A recent study showed that genotype C was the dominant strain in Inner Mongolia [12]; however, the prevalence and molecular characteristics of BPIV3 in China remain largely unknown.

To better understand the epidemiology and genomic characteristics of BPIV3, we have systematically surveyed BPIV3 prevalence in China and analyzed its molecular features. In this study, a total of 776 respiratory samples of beef cattle and yak were collected from 16 provinces and one municipality in China between September 2020 and June 2022 to detect the prevalence of BPIV3. Moreover, 22 HN gene and 9 near-complete genome sequences were obtained from BPIV3-positive samples and analyzed, laying the foundation for further studies on genetic variation of BPIV3 genotype C strains in China. The findings help improve our understanding of the evolution and molecular characteristics of BPIV3.

## 2. Materials and Methods

### 2.1. Sample Collection

From September 2020 to June 2022, a total of 776 respiratory samples were collected from beef cattle and yak at 2–6 months of age and with BRDC from 16 provinces and one municipality in China (Table 1). The typical clinical characteristics of beef cattle and yak with BRDC are a runny nose, cough, and dyspnea. All samples were transported on ice to the laboratory and stored at −80 °C.

### 2.2. RNA Extraction and cDNA Synthesis

The nasal swabs were placed in 1 mL phosphate buffer saline (PBS), vigorously vortexed, and centrifuged at 10,000× *g* for 10 min. The lung tissues were fully ground, and the homogenates were diluted 1:3 (*w*/*v*) with PBS. Then, the samples were prepared by freeze-thawing three times, and the lysed samples were centrifuged (10,000× *g*, 10 min) and the supernatants collected. According to the kit instructions, viral RNA was extracted from 300 µL of the sample suspension using a Nucleic Acid Co-Prep Kit (GeneRadar, Xiamen, China), and a PrimeScript RT Reagent Kit (TaKaRa, Kyoto, Japan) was used to synthesize cDNA, which was then stored at −20 °C.

### 2.3. Detection of BPIV3

A total of 776 respiratory samples were screened for BPIV3 detection by a specific reverse transcription insulated isothermal PCR (RT-iiPCR) assay established at our laboratory. A POCKIT^TM^ device (GeneRadar, Xiamen, China) was used for BPIV3 detection, where the program was run with the default parameters, and the tubes were incubated at 95 °C for 58 min to complete the reaction. The sequences of primers used were the following: F: 5′-ARAGGACACAGAAGAGAGCACT-3′; R: 5′-TRGCCACACATACAACTCTCT-3′, and the sequence of the probe was FAM-TTACAGAAAGGGCGATTACATTATTACAGA-BHQ1 (targeting the P gene; amplification length 125 bp). The reaction system was as follows: 3 μL forward primer (10 μM), 3 μL reverse primer (10 μM), 0.25 μL probe (10 μM), 2 μL cDNA, 16.75 μL double distilled water, and 25 μL Premix Ex Taq^TM^ (Probe qPCR) Kit (5 U/μL) (TaKaRa, Dalian, China).

### 2.4. Amplification of HN and Genome Sequences

Based on known BPIV3 genomes, a pair of primers (HN-F: 5′-CGACAATAGCAATGAACCC-3′; HN-R:5′-TGTTGATGCCTGTCTTCTGT-3′) was designed to amplify the complete HN gene sequences. Subsequently, a panel of primers was designed for BPIV3 genome amplification from BPIV3-positive samples (Table 2). PCR products were purified and cloned into the pMD™ 19T-vector (TaKaRa, Dalian, China) for sequencing.

### 2.5. Sequence, Phylogenetic, and Recombination Analysis

Using the MegAlign program of DNASTAR 7.0 software (DNASTAR Inc., Madison, WI, USA), the homologies of the nucleotide and amino acid sequences were determined. Utilizing SeqMan software (version 7.0; DNA Star), the gene sequences were assembled. A phylogenetic tree was constructed with MEGA version 7.0 using the maximum likelihood (ML) method. The Bayesian phylogenetic tree was constructed using MrBayes software. Recombination analyses were conducted using the RDP4 program [13,14]. The AlphaFold v2.0 was used to predict the protein structure [15], and the protein domains were predicted by SMART (http://smart.embl-heidelberg.de/), accessed on 7 February 2023.

## 3. Results

### 3.1. Detection of BPIV3

For the total sample, the detection rate of BPIV3 was 18.17% (141/776). Of the beef cattle nasal swab samples, 107 of the 645 (16.59%) samples were detected as BPIV3-positive by RT-iiPCR and distributed in six provinces (Figure 1): Sichuan (60/173, 34.68%), Ningxia (4/30, 13.33%), Inner Mongolia (16/58, 27.59%), Hebei (12/61, 19.67%), Shanxi (12/52, 23.08%), and Jilin (3/10, 30.00%). The overall positive rate of BPIV3 in the yak samples was 25.59% (34/131). The detection rates of BPIV3 in nasal swab samples and lung tissue samples from yaks in Sichuan were 16.90% (12/71) and 44.00% (22/50), respectively (Table 1).

### 3.2. Bioinformatics Analysis of HN Gene Sequences

A total of 22 HN gene sequences (GenBank numbers: OP908132–OP908153) were amplified from BPIV3-positive samples from 13 farms in 6 provinces. Among them, 19 HN genes and the 3 HN genes were amplified from beef cattle and yak, respectively. The 22 HN gene sequences had a length of 1719 bp and encoded 573 amino acids (aa). The ML phylogenetic tree, which was constructed using all available complete HN gene sequences of BPIV3 in GenBank, revealed that the 22 strains belonged to genotype C and clustered with sequences from other Chinese genotype C strains. Furthermore, all overseas BPIV3 genotype C strains clustered into distinct clusters (Figure 2). Multiple sequence alignments indicated that the HN gene sequences identified in this study displayed an 81.10–99.80% nucleotide (nt) identity and 97.40–99.80% aa sequence similarity with all 61 BPIV3 genotype C complete HN gene sequences in GenBank. Further sequence analysis revealed that, compared to the HN gene sequences of the 61 available BPIV3 genotype C strains from GenBank, the Chinese strain HN sequences had a unique aa mutation (T84A). Structural modeling of the HN protein showed that the mutation was located in the α-helix and the predicted structural domain of the HN protein ranged from 34–571 aa and had only one transmembrane region, which was located at 34–56 aa (Figure 3).

### 3.3. Bioinformatics Analysis of the Complete Genomes

Nine near-complete genomes were obtained from the clinical samples (GenBank numbers: OP718792–OP718797, OM621819, OM782290, and OM782291), which were all 15,474 bp in length. Among them, six genomes and three genomes were amplified from beef cattle and yak, respectively. All the complete BPIV3 sequences were used to create a Bayesian phylogenetic tree that showed the nine strains belonged to genotype C and clustered with all the Chinese strains in one large clade, while the seven overseas strains of BPIV3 genotype C clustered into another clade (Figure 4). Multiple sequence alignments indicated that the nine near-complete genomes shared 99.2%–99.8% nt identity (99.3–99.8% aa identity) with one another and shared 97.4–99.8% nt identity (97.5–99.8% aa identity) with all the complete genomes of overseas BPIV3 genotype C strains in GenBank. Meanwhile, no recombination events were found in any of them. Fascinatingly, all the Chinese strains (including the nine strains in this study and others from the uploaded three strains) shared 45 nt mutations (Table 3), which were not demonstrated by the seven known genomes of overseas BPIV3 genotype C strains. Five mutations occurred in non-coding regions, accompanied by thirty-five nonsense mutations and five sense mutations, which resulted in aa mutations in the N (P439L, T494A), F (N452D, M497I), and HN (T84A) proteins. 

## 4. Discussion 

BRDC is a multifactorial disease complex of cattle that restricts the healthy development of cattle breeding worldwide [16]; however, the prevalence and molecular features of BPIV3, one of the major causative agents of BRDC, are still mostly unknown in most Chinese provinces. The findings of this study showed that 16.59% (107/645) of the beef cattle nasal swab samples were positive for BPIV3, and the positive samples were scattered over 14 farms located in 6 provinces. The reason for this may be the rapid development of the cattle industry in these provinces in recent years and the frequent transportation of live cattle, thus leading to the spread of the virus. In addition, the geographic distances between sampling localities were large (in some cases over 2500 km), which indicates that BPIV3 has a broad geographical distribution in China. We previously demonstrated the presence of BPIV3 in yak, but the prevalence remains unknown [17]. In this study, 25.95% (34/131) of yak respiratory samples were detected as BPIV3-positive, which suggests that BPIV3 is widely present in yak. Yak (*Bos grunniens*) are members of the family *Bovidae* and genus *Bos* [18],. There are more than 14 million yak worldwide, and they are found at high altitudes (above 2500–6000 m) in China, India, Nepal, Pakistan, Kyrgyzstan, Mongolia, and Russia [19], representing the main income source for herdsmen in plateau areas [20]. The finding will aid in yak respiratory disease diagnosis and prevention. In addition, BPIV3 genotype C was found to be the dominant BPIV3 strain in cattle herds in China in the present study, with some unique genetic characteristics. These findings help improve our understanding of the evolution and molecular characteristics of BPIV3.

Based on the genome sequence analysis, the Chinese BPIV3 genotype C strain shares five unique aa mutations in the N (P439L, T494A), F (N452D, M497I), and HN (T84A) proteins. The N protein in *Paramyxovirinae* is highly conserved and immunogenic. Both ends of the BPIV3 N protein (aa 9–157, N-terminal; aa 391–500, C-terminal) were identified to have good immunogenicity, and the C-terminal domain epitopes can be used to develop a diagnostic analysis that can discriminate between different genotypes [21]. Fascinatingly, comparing to the N gene sequences of BPIV3, the aa mutations (P439L, T494A) were found in all Chinese BPIV3 genotype C strains, and these mutations were also found in genotype A strains. This phenomenon is significant as it may add to the difficulties in developing diagnostic methods that differentiate between genotypes of BPIV3. In addition, sequence comparison of the viral genomes with known paramyxoviruses showed that BPIV3 and human parainfluenza type 3 virus (HPIV3) were the most similar [6]. In the study of HPIV3, an immunodominant region was also found in the C-terminal 397–486 region of N protein, and the above antigenic sites on N protein were found to be important in triggering the humoral immune response against HPIV3 [22]. This indicates that further study of the biological significance of the N protein mutation sites (P439L, T494A) of BPIV3 genotype C strains in China should help better diagnose, prevent, and control BRDC.

HN and F are two glycoproteins on the viral envelope, which play a crucial role in virus entry into host cells [23]. The paramyxovirus F protein belongs to the class I viral fusion protein type and is produced as inactive precursors (F0), which are then split into two disulfide-linked domains during cellular processing (F1 and F2) [24]. Currently, there is a paucity of research on the BPIV3 F0 protein, but there are many studies on a very similar HPIV3, and this information can provide a reference for future studies on the BPIV3 F0 protein. For instance, in the secondary structure of HPIV3, F1 is mainly composed of HRA (129–192aa) and HRB (447–484aa), of domains DIII (193–284aa), DI (285–368aa), and DII (375–428aa) [25,26]. In this study, it was found that one of the two mutation sites of the F protein of BPIV3 (N452D) may be located in the HRB junction region of F1. Related research has shown that this HRB structure has a significant impact on virus and host cell fusion performance [27]. For example, in the NDV study, four amino acids (L436, E439, I450, and S453) in the HRB region were identified, which can regulate the fusion ability or the expression of the active form to a certain extent [28]. In HPIV3, T429 and N446 were identified as the key amino acids of the HRB region affecting the fusion process [25]. Additionally, several studies indicated that point mutations in the F protein might impact the virus virulence [27]. Therefore, the effects of F protein mutations in BPIV3 genotype C of the Chinese strain on the function of the HRB region and viral virulence are worthy of further study.

HN glycoprotein is a multifunctional protein that induces efficient membrane fusion of the virus with the host cell and mediates virus entry into the host cell through a complex mechanism [24]. The predicted structure of the HN protein showed that the mutation at 84 sites changed from alanine to threonine. Alanine was a hydrophobic aa and threonine was a hydrophilic aa. A hydrophobic aa is generally located within proteins and plays an important role in maintaining the protein structure and antibody–antigen interactions; further studies are needed to determine the effect of mutations on the viral protein structure and function. In addition, previous studies showed that aa mutations in HN protein affect the formation of binding sites to sialic-acid-containing receptors, for instance, the loss of the binding site caused by the Arg516 mutation in NDV HN significantly reduced the fusion properties and growth rate of the virus, demonstrating that the binding site on the NDV HN protein is closely related to virus infection and replication [29]. Moreover, previous findings indicated antigenicity differences between BPIV3 isolates, mainly in the epitope envelope glycoprotein HN, differences that are related to their pathogenicity and virulence in animals [30]. Meanwhile, in a study of the HN protein of NDV in *Paramyxoviridae*, it was found that a single nucleotide mutation in the HN protein either made the virus a highly pathogenic strain or reduced the virulence of the virus [31,32]. Therefore, the effects of the unique mutation of the HN protein in BPIV3 genotype C of the Chinese strain on the fusion performance, pathogenicity, and virulence of the virus must be further studied.

## 5. Conclusions

In this study, we detected the prevalence of BPIV3 on beef cattle and yak farms in different regions of China and further analyzed the genetic evolution of the HN gene and genome of BPIV3 strains. We confirmed that BPIV3 genotype C strains, the dominant strains in China, have a broad geographical distribution, and the Chinese BPIV3 genotype C strains considered showed some unique genetic characteristics, which improve our knowledge of the prevalence and genetic variation of BPIV3.

## Figures and Tables

**Figure 1 animals-13-00793-f001:**
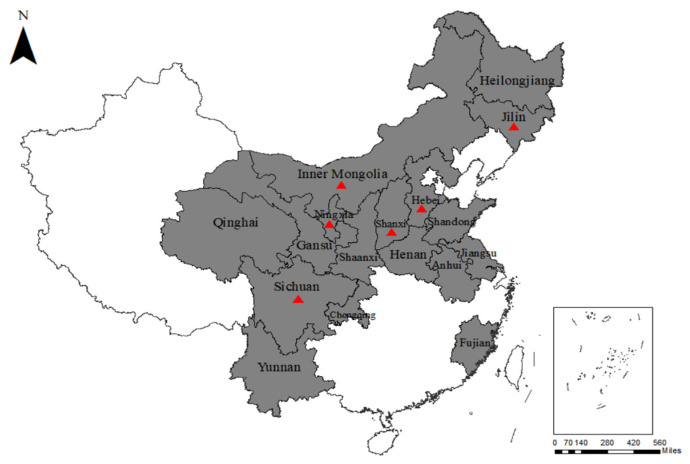
Map of China displaying the areas of sample collection. The 17 sampled regions are indicated in gray. The red triangles (
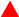
) represent the sampling areas that tested positive for BPIV3.

**Figure 2 animals-13-00793-f002:**
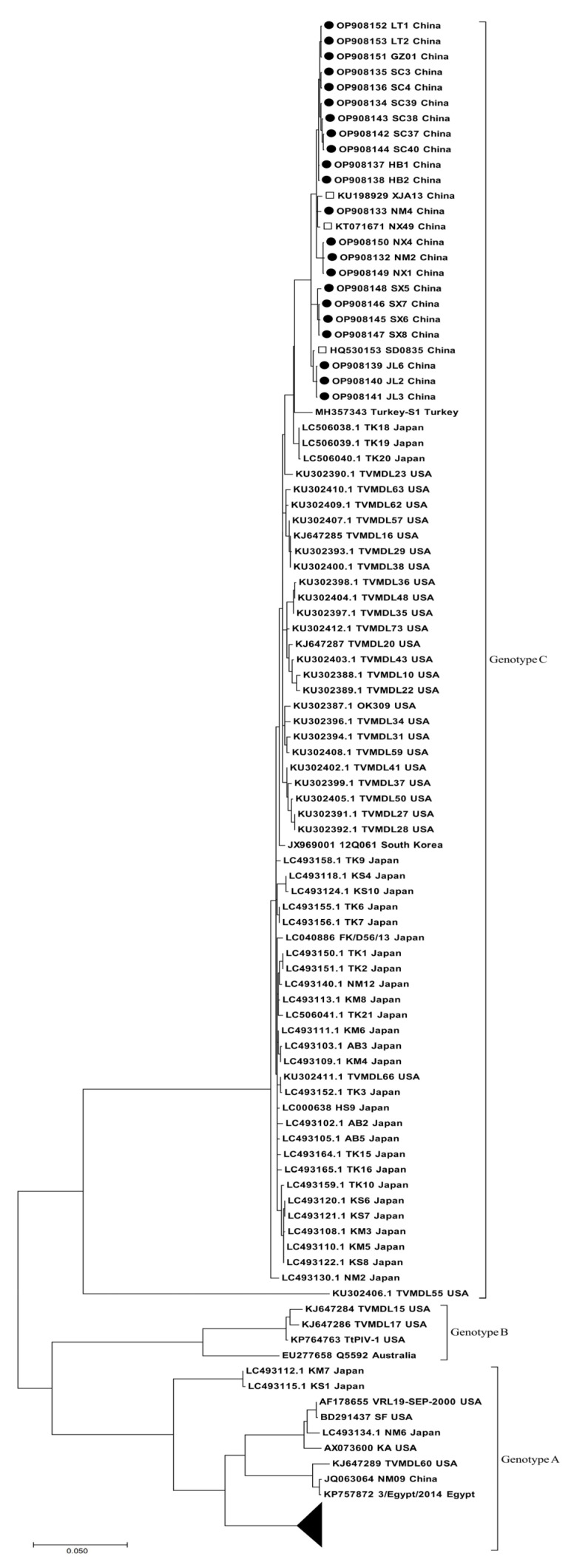
Phylogenetic tree based on HN gene sequences. MEGA 7.0 software was used to conduct multiple sequence alignments, and a phylogenetic tree was created using the ML method. The black circular marker sequences were obtained by amplification in this study; the black box markers are the sequences of other genotype C strains in China; the black triangle represents the set of HN sequences of BPIV3 genotype A sequences that are not shown in the phylogenetic tree.

**Figure 3 animals-13-00793-f003:**
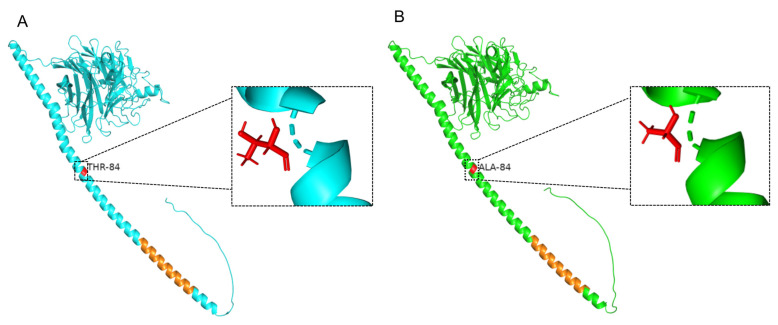
Structural model of HN protein. (**A**) The sequences from this study (GenBank number: OP908143). (**B**) Sequence as a reference (GenBank number: MH357343). Red bars represent mutation sites and orange bars represent transmembrane domains.

**Figure 4 animals-13-00793-f004:**
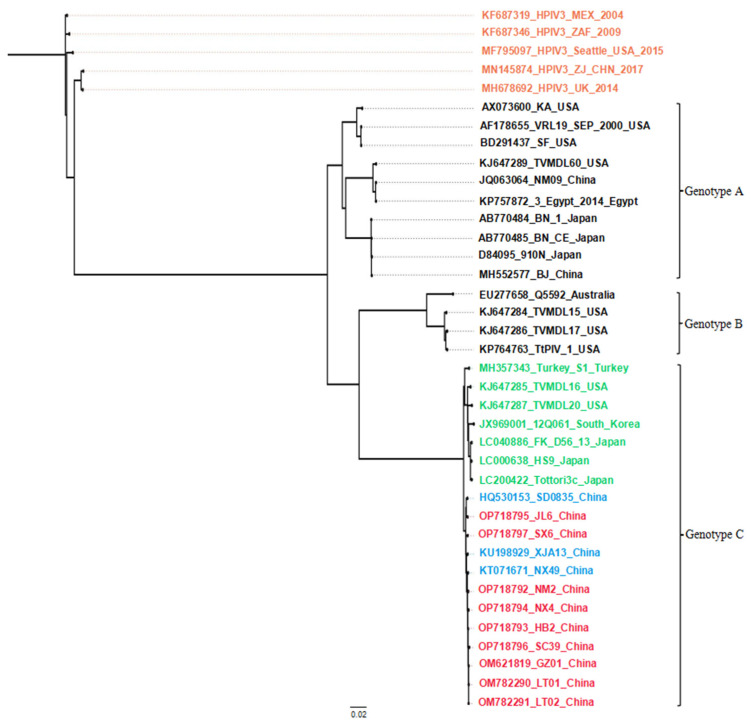
Phylogenetic tree based on complete genomes. A phylogenetic tree was created using the MrBayes software. The orange font represents human parainfluenza virus type 3 sequences; the green font represents overseas BPIV3 genotype C strains; the blue font represents BPIV3 genotype C strains uploaded by others in China; and the red font represents the Chinese strain sequences in this study.

**Table 1 animals-13-00793-t001:** Prevalence of BPIV3 in respiratory samples from different regions of China.

Region	Sample Source	Sample Type	Number of Farms	Number of Samples	Positive Rate (%)
Sichuan	Beef cattle	Nasal swab	10	173	34.68% (60/173)
Fujian	Beef cattle	Nasal swab	1	10	0.00% (0/10)
Ningxia	Beef cattle	Nasal swab	2	30	13.33% (4/30)
Inner Mongolia	Beef cattle	Nasal swab	5	58	27.59% (16/58)
Jiangsu	Beef cattle	Nasal swab	1	15	0.00% (0/15)
Henan	Beef cattle	Nasal swab	4	40	0.00% (0/40)
Hebei	Beef cattle	Nasal swab	4	61	19.67% (12/61)
Shanxi	Beef cattle	Nasal swab	4	52	23.08% (12/52)
Chongqing	Beef cattle	Nasal swab	4	83	0.00% (0/83)
Shandong	Beef cattle	Nasal swab	1	30	0.00% (0/30)
Jilin	Beef cattle	Nasal swab	1	10	30.00% (3/10)
Qinghai	Beef cattle	Nasal swab	2	16	0.00% (0/16)
Anhui	Beef cattle	Nasal swab	1	15	0.00% (0/15)
Yunnan	Beef cattle	Nasal swab	1	10	0.00% (0/10)
Heilongjiang	Beef cattle	Nasal swab	1	12	0.00% (0/12)
Shaanxi	Beef cattle	Nasal swab	1	10	0.00% (0/10)
Gansu	Beef cattle	Nasal swab	2	20	0.00% (0/20)
Sichuan	Yak	Nasal swab	7	71	16.9% (12/71)
Qinghai	Yak	Nasal swab	1	10	0.00% (0/10)
Sichuan	Yak	Lung tissue	5	50	44.00% (22/50)

**Table 2 animals-13-00793-t002:** PCR primers for complete genome amplification of BPIV3.

Name	Primer Sequence (5′-3′)	Position
BPIV3-1F	ACCAAACAAGAGGAGAGACTTG	1–1963
BPIV3-1R	ATTGTTGTGCTGAGCCTTGT	
BPIV3-2F	AGACTCCATCCACAACCCA	1750–3697
BPIV3-2R	CTTGTGTCTGGGAACTACTGTG	
BPIV3-3F	TCAAAGGCAAAACAGTCATACAT	3482–5728
BPIV3-3R	TCCTACTGAGCTTTGAATTGACTGT	
BPIV3-4F	AACAGTACTAGTTCCAGGAAGAAGC	5396–7884
BPIV3-4R	GGACAGCCAGTTAAATTGCATATTAC	
BPIV3-5F	CAGTAGGACCGGGGATTTA	7880–9902
BPIV3-5R	TGTCCTCCGTGTCTTTCTCTA	
BPIV3-6F	CCTTTTTCCGAACTTTTGG	9734–12,509
BPIV3-6R	GTAGTCACTGGTGTCAGAATCTTTA	
BPIV3-7F	AGGAGGAAGAATGATAAATGG	12,105–13,822
BPIV3-7R	TTCTTGGATTATCGTCACAGTTA	
BPIV3-8F	GTGTGTTGTTTAGCAGAAATAGC	13,501–15,474
BPIV3-8R	ACCAAACAAGAGAAAAACTCTGT	

**Table 3 animals-13-00793-t003:** Nucleotide and amino acid differences between the complete genomes of the Chinese strains in this study and all BPIV3 genotype C overseas strains.

Nucleotide Position	Nucleotide Mutations		Amino Acid Mutations	
	Chinese Strains	Overseas Strains	Chinese Strains	Overseas Strains
1088	T	C	-	-
1169	A	G	-	-
1355	T	C	-	-
1426	C	T	P	L
1436	C	T	-	-
1590	A	G	T	A
1666	A	G	/	/
3659	A	C	/	/
4096	T	C	-	-
4234	C	T	-	-
4357	C	T	-	-
4603	T	C	-	-
4795	A	G	-	-
4938	A	G	/	/
4968	A	C	/	/
5654	T	C	-	-
6474	A	G	N	D
6611	G	A	M	I
6767	A	G	/	/
7082	C	T	-	-
7104	A	G	T	A
7310	T	C	-	-
7847	A	G	-	-
7916	C	T	-	-
7943	C	T	-	-
7946	C	T	-	-
7952	C	T	-	-
7991	T	C	-	-
8180	T	C	-	-
8282	A	G	-	-
8453	G	A	-	-
8701	A	C	/	/
9424	C	T	-	-
9784	C	T	-	-
10,342	A	G	-	-
11,815	T	C	-	-
12,082	A	G	-	-
12,853	T	C	-	-
13,853	C	T	-	-
14,261	C	T	-	-
14,367	T	C	-	-
14,607	C	A	-	-
14,637	A	G	-	-
14,904	G	A	-	-
14,961	A	C	-	-

Note: - nonsense mutation; / noncoding region.

## Data Availability

The HN gene sequences and near-complete genomes were uploaded to GenBank (GenBank numbers: OP908132-OP908153, OP718792-OP718797, OM621819, OM782290, and OM782291).

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
