# Peer review of "Prevalence and Molecular Characterization of Bovine Parainfluenza Virus Type 3 in Cattle Herds in China"

_animals, 2023, doi:10.3390/ani13050793_

Round 1

Reviewer 1 Report

The present study aimed at the study was to identify the prevalence and genetically characterize Bovine parainfluenza virus type 3 in beef cattle and yaks in China.

 This study is interesting and I consider that it may be suitable for publication in the journal Animals.

 I only have a few minor corrections for this study.

I suggest authors include other keywords that are not included in the title of the manuscript.

The Neighbor-Joining (NJ) method is not an adequate phylogenetic method, for several reasons, mentioning some of which the NJ method does not take into account the ancestor-descendant relationship. The trees obtained using NJ do not reflect evolutionary relationships, in reality, they are groupings of very similar members that cannot be considered clades or monophyletic groups. I suggest that you use other methods to build your phylogenetic trees than in the MEGA program you can find other alternatives.

Author Response

Response to Reviewer 1 Comments

Dear Reviewer:

Thank you for kindly reviewing the manuscript entitled “Prevalence and Molecular Characterization of Bovine Parainfluenza Virus Type 3 in cattle herds in China” (animals-2175528). We are grateful for the professional criticisms and suggestions of the reviewers. All issues mentioned in the reviewers' comments have been revised accordingly and revisions are marked in red typeface in the revised manuscript.

Point 1: I suggest authors include other keywords that are not included in the title of the manuscript.

Response 1: Thanks for your professional suggestions. We have now added some keywords according to your professional suggestions in the revised manuscript.

Point 1: The Neighbor-Joining (NJ) method is not an adequate phylogenetic method, for several reasons, mentioning some of which the NJ method does not take into account the ancestor-descendant relationship. The trees obtained using NJ do not reflect evolutionary relationships, in reality, they are groupings of very similar members that cannot be considered clades or monophyletic groups. I suggest that you use other methods to build your phylogenetic trees than in the MEGA program you can find other alternatives.

Response 2: Thanks for your professional suggestions. We strongly agree with you that we have added a Bayesian phylogenetic tree to the Fig 4 to better show the evolutionary relationships. Meanwhile, we constructed a phylogenetic tree of HN genes using Bayesian analysis, but the typing results of BPIV3 were not straightforward. In addition, considering that the BPIV3 genotype C strain was originally identified based on the HN gene, using the NJ method to construct a phylogenetic tree [1], and that a large number of researchers have continued to use the NJ method as a quick and easy typing method for BPIV3[1-4], we chose to keep the HN gene phylogenetic tree constructed by the NJ method for typing, meanwhile, we revised the description of the NJ phylogenetic tree in the results section to make the results more rigorous. Thank you again for your professional advice and we sincerely hope to get your understanding.

  1. Zhu, Y.M.; Shi, H.F.; Gao, Y.R.; Xin, J.Q.; Liu, N.H.; Xiang, W.H.; Ren, X.G.; Feng, J.K.; Zhao, L.P.; Xue, F. Isolation and genetic characterization of bovine parainfluenza virus type 3 from cattle in China. Vet Microbiol 2011, 149, 446-451, doi:10.1016/j.vetmic.2010.11.011.
  2. Horwood, P.F.; Gravel, J.L.; Mahony, T.J. Identification of two distinct bovine parainfluenza virus type 3 genotypes. J Gen Virol 2008, 89, 1643-1648, doi:10.1099/vir.0.2008/000026-0.
  3. Maidana, S.; Lomonaco, M.; Odeon, A.; Combessies, G.; Graig, M.I.; Rodriguez, D.; Parenno, V.; Zabal, O.; Konrad, J.L.; Crudelli, G. Isolation and characterization of bovine parainfluenza virus type 3 from water buffaloes (Bubalus bulalis) in Argentina. Revista Colombiana De Psiquiatria 2012, 8, 249-260.
  4. Oem, J.K.; Lee, E.Y.; Lee, K.K.; Kim, S.H.; Lee, M.H.; Hyun, B.H. Molecular characterization of a Korean bovine parainfluenza virus type 3 isolate. Vet Microbiol 2013, 162, 224-227, doi:10.1016/j.vetmic.2012.10.013.

Reviewer 2 Report

In this study, the authors have investigated and characterized the prevalence of Bovine parainfluenza virus type 3 (BPIV3) in China between September 2020 and June 2022. They have found that 18.17% (141/776) samples (from 6 different provinces) are positive for BPIV3 by RT-iiPCR assay. Moreover, 22 complete HN gene sequences and 9 nearly complete genomes sequences were obtained from these positive samples, and they are all clustered into genotype C based on the phylogenetic analysis. Though the authors' meaning is clear, the manuscript needs correction by a native English speaker because there are grammatical problems that I did not correct. In general, the study is well done, except for some concerns as outlined for the authors to consider below.

Major comments:

1, Results, lines 114-126, the description on detection rates of BPIV3 in different provinces of China should be modified. Besides simply presenting the data,  the authors should analyze them and make conclusions accordingly.

2, Fig. 1 could be improved to clearly present the results on detection of BPIV3 in different provinces of China.

3, Results, lines 142-143, please provide the evidence for the statement “…, the Chinese strains HN sequences had unique aa mutation 142 (T84A)”. It would be better to display this position in a protein structure model and discuss it accordingly.

4, Please deeply analyzed the genetic data presented in this study and emphasize the evidences in the manuscript to support the conclusion that Chinese BPIV3 strains has a unique evolutionary trend.

Minor comments:

5, Abstract, line 21, please provide the full name of RT-iiPCR.

6, Please try the maximum likelihood method in the phylogenetic analysis of this study.

7, What do black triangles in the Fig. 2 represent?

8, Table 2, please also indicate the amino acid differences corresponding to the nucleotide differences between BPIV3 Chinses strains and oversea strains.

9, There are many typos and grammatical mistakes in the manuscript. For example, “…different provinces strains” in line 22, “…for BPIV3-positive” in line 23, “…sequences were obtained from 6 provinces” in line 25, “…the antibody-positive rate…” in line 48, please correct them accordingly.

10, Please change or rephrase following sentences to make them correct in grammar, rigorous in description and clear in meaning.

(1) lines 32-33, “These findings will aid in understanding of the evolution and molecular characteristics of BPIV3.”

(2) lines 37-41, the sentence is too long to read smoothly.

(3) line 47, “Nowadays, BPIV3 three genotypes were found in China.”

(4) lines 52-53, “…whereas there is limited information on prevalent genotypes of BPIV3, genomic features, and molecular epidemiological investigations in most parts of China.”

(5) lines 134-136, “The 129 HN gene sequences were used to create a phylogenetic tree that showed the 22 strains belonged to genotype C and clustered in one large clade with all Chinese strains.”

Author Response

Response to Reviewer 2 Comments

Dear Reviewer:

Thank you for kindly reviewing the manuscript entitled “Prevalence and Molecular Characterization of Bovine Parainfluenza Virus Type 3 in cattle herds in China” (animals-2175528). We are grateful for the professional criticisms and suggestions of the reviewers. We apologize for the poor English and we have tried our best to polish the English/grammar, and then asked a native English speaker professional to proof read the paper.. All issues mentioned in the reviewers' comments have been revised accordingly and revisions are marked in red typeface in the revised manuscript.

Major comments:

Point 1: Results, lines 114-126, the description on detection rates of BPIV3 in different provinces of China should be modified. Besides simply presenting the data, the authors should analyze them and make conclusions accordingly.

Response 1: Thanks for your professional suggestions. We have revised the description of detection rates of BPIV3 in different provinces of China, which made the results more concise and clearer, the results are analyzed in the revised manuscript.

Point 2: Fig. 1 could be improved to clearly present the results on detection of BPIV3 in different provinces of China.

Response 2: We have correspondingly improved the presentation of the results on the detection of BPIV3 in different provinces of China in Figure 1.  

Point 3: Results, lines 142-143, please provide the evidence for the statement “…, the Chinese strains HN sequences had unique aa mutation 142 (T84A)”. It would be better to display this position in a protein structure model and discuss it accordingly.

Response 3: Thanks for your professional suggestions. We have added a structural model of the HN protein (Figure 3) to the revised manuscript to show the amino acid mutations at site 84 in this study, and added a discussion of the corresponding section.

Point 4: Please deeply analyzed the genetic data presented in this study and emphasize the evidences in the manuscript to support the conclusion that Chinese BPIV3 strains has a unique evolutionary trend.

Response 4: Thanks for your professional suggestions. We have added a Bayesian phylogenetic tree in the revised manuscript to better analyze and describe the genetic evolutionary relationships of the Chinese BPIV3 genotype C strains, which provides stronger evidence that the Chinese BPIV3 genotype C strain has a unique evolutionary trend.

Minor comments:

Point 5: Abstract, line 21, please provide the full name of RT-iiPCR.

Response 5: Thank you for your advice, we have provided the full name of RT-iiPCR in the revised manuscript.

Point 6: Please try the maximum likelihood method in the phylogenetic analysis of this study.

Response 6: We constructed the phylogenetic tree using the maximum likelihood (ML) method, which is not significantly different from the phylogenetic tree constructed by the neighbor joining (NJ) method (shown below), and since another reviewer strongly suggested changing the method of phylogenetic tree construction and suggested that we do not use the MEGA software program to construct the phylogenetic tree, we chose to use the Bayesian phylogenetic tree. We ask for your understanding.

Point 7: What do black triangles in the Fig. 2 represent?

Response 7: We feel sorry about this mistake, the black triangle in Figure 2 represents the sequence not shown in the phylogenetic tree and we have revised this issue in the revised manuscript.

Point 8: Table 2, please also indicate the amino acid differences corresponding to the nucleotide differences between BPIV3 Chinses strains and oversea strains.

Response 8: We have revised the table according to the request of the reviewers.

Point 9: There are many typos and grammatical mistakes in the manuscript. For example, “…different provinces strains” in line 22, “…for BPIV3-positive” in line 23, “…sequences were obtained from 6 provinces” in line 25, “…the antibody-positive rate…” in line 48, please correct them accordingly.

Response 9: We feel sorry about these mistakes, and we have corrected it accordingly in the revised manuscript.

Point 10: Please change or rephrase following sentences to make them correct in grammar, rigorous in description and clear in meaning.

(1) lines 32-33, “These findings will aid in understanding of the evolution and molecular characteristics of BPIV3.”

(2) lines 37-41, the sentence is too long to read smoothly.

(3) line 47, “Nowadays, BPIV3 three genotypes were found in China.”

(4) lines 52-53, “…whereas there is limited information on prevalent genotypes of BPIV3, genomic features, and molecular epidemiological investigations in most parts of China.

(5) lines 134-136, “The 129 HN gene sequences were used to create a phylogenetic tree that showed the 22 strains belonged to genotype C and clustered in one large clade with all Chinese strains.”

Response 10: Thanks for your professional suggestions and we have revised these sentences in the manuscript accordingly.

(1) lines 32-33, the sentence has been changed to “These findings will contribute to understanding of the epidemiological characteristics and genetic evolution of BPIV3 in China.“

(2) lines 37-41, we have revised this sentence to avoid too long to read smoothly.

(3) line 47, the sentence has been changed to “So far, three BPIV3 genotypes have been found in China.“

(4) lines 52-53, the sentence has been revised to “…however, the prevalence and molecular characteristics of BPIV3 in China remain largely unknown.

(5) lines 134-136, we have revised this sentence.

Reviewer 3 Report

In this manuscript, 141 BPIV3 positive samples were detected from 21 farms in 6 provinces in China.  HN genes and viral genomes were sequenced, and all strains were clustered into genotype C. These results indicated genotype C BPIV3 was prevalence in cattle herds in China.

My comments to the authors are as follows(minor):

In section 3.4, sequence comparison displayed 45 site mutations in Table 2. The discussion focused on the analysis of HN and F proteins which were related to the virulence (F) and the adsorption capacity (HN). The cleavage site of F0 and the transmembrane domain of HN should be showed. Further more, comparison of protein structure prediction results can also support the discussion.

Author Response

Response to Reviewer 3 Comments

Dear Reviewer:

Thank you for kindly reviewing the manuscript entitled “Prevalence and Molecular Characterization of Bovine Parainfluenza Virus Type 3 in cattle herds in China” (animals-2175528). We are grateful for the professional criticisms and suggestions of the reviewers. All issues mentioned in the reviewers' comments have been revised accordingly and revisions are marked in red typeface in the revised manuscript.

Point 1:  In section 3.4, sequence comparison displayed 45 site mutations in Table 2. The discussion focused on the analysis of HN and F proteins which were related to the virulence (F) and the adsorption capacity (HN). The cleavage site of F0 and the transmembrane domain of HN should be showed. Furthermore, comparison of protein structure prediction results can also support the discussion

Response 1: Thanks for your professional suggestions. We have added the predicted structural model of HN protein and its transmembrane domain in the revised manuscript, unfortunately, we have viewed a large number of related articles, and have not seen the report on the F0 cleavage site of BPIV3, and we found that the F0 cleavage site of different members of Paramyxoviridae is very different. For example, the F0 cleavage site of Newcastle disease virus in Paramyxoviridae is 112-117 aa [1], but the F0 cleavage site of Nipah virus in the same Paramyxoviridae is widely distributed in F protein [2,3], so we cannot find the F0 cleavage site of BPIV3. In addition, we have also revised the discussion in this section to make the results clearer and more rigorous. We sincerely hope to get your understanding.

  1. Dutch, R.E.; Hagglund, R.N.; Nagel, M.A.; Paterson, R.G.; Lamb, R.A. Paramyxovirus fusion (F) protein: a conformational change on cleavage activation. Virology 2001, 281, 138-150, doi:10.1006/viro.2000.0817.
  2. Moll, M.; Diederich, S.; Klenk, H.D.; Czub, M.; Maisner, A. Ubiquitous activation of the Nipah virus fusion protein does not require a basic amino acid at the cleavage site. J Virol 2004, 78, 9705-9712, doi:10.1128/JVI.78.18.9705-9712.2004.
  3. Diederich, S.; Moll, M.; Klenk, H.D.; Maisner, A. The nipah virus fusion protein is cleaved within the endosomal compartment. J Biol Chem 2005, 280, 29899-29903, doi:10.1074/jbc.M504598200.

Round 2

Reviewer 2 Report

In this revised manuscript, the authors have addressed some of reviewer's questions. However, some additional concerns need to be considered by the authors. In addition, there are still some typos and grammatical mistakes in the manuscript. Please proofread carefully and make changes accordingly.

Major comments:

1, I suggest the authors to replace the NJ phylogenetic tree with the ML phylogenetic tree in the manuscript. NJ method is faster in term of computing but quite weaker compare to ML method. ML method uses more complex evolution model, and the mathematics behind ML or BI methods are known to be stronger than NJ for reconstructing sequence histories.

2, In the phylogenetic tree, some sets of sequences are not shown (marked as the black triangles in Fig. 2). What are the sequences? In the text, the authors state that “…, all overseas strains of BPIV3 genotype C clustered into another cluster (Fig. 2)”. Are all the sequences compacted within the black triangle from oversea strains of BPIV3 genotype C? Based on the ML phylogenetic tree in the responses, the statement “all overseas strains of BPIV3 genotype C clustered into another cluster (Fig. 2)” is less rigorous. Since there are only 61 available HN gene sequences of BPIV3 genotype C, it would be better to generate the phylogenetic tree figure showing all these sequences with the compact arrangement.

3, Results, lines 159-161, “Structural modeling of the HN protein shows that the mutation was located in the α-helix and the HN protein was predicted to contain one transmembrane region, which was located at 34–56 aa (Fig.3)”. Which domain/part is the mutation/α-helix located or near to? Why the authors mention the transmembrane region here?

4, In this revised version of the manuscript, though the authors have found some unique genetic characteristics in sequences of Chinese BPIV3 genotype C strains, the analyses and evidences to support the conclusion that Chinese BPIV3 strains has a unique evolutionary trend is inadequate.

Author Response

Response to Reviewer 2 Comments

Dear Reviewer:

Thank you for kindly reviewing the manuscript entitled “Prevalence and Molecular Characterization of Bovine Parainfluenza Virus Type 3 in cattle herds in China” (animals-2175528). We are grateful for the professional criticisms and suggestions of the reviewers. We have tried our best to polish the English/grammar. All issues mentioned in the reviewers' comments have been revised accordingly and revisions are marked in red typeface in the revised manuscript.

Major comments:

Point 1: I suggest the authors to replace the NJ phylogenetic tree with the ML phylogenetic tree in the manuscript. NJ method is faster in term of computing but quite weaker compare to ML method. ML method uses more complex evolution model, and the mathematics behind ML or BI methods are known to be stronger than NJ for reconstructing sequence histories.

Response 1: Thanks for your professional suggestions. We have replaced NJ phylogenetic tree with the ML phylogenetic tree in the manuscript.

Point 2: In the phylogenetic tree, some sets of sequences are not shown (marked as the black triangles in Fig. 2). What are the sequences? In the text, the authors state that “…, all overseas strains of BPIV3 genotype C clustered into another cluster (Fig. 2)”. Are all the sequences compacted within the black triangle from oversea strains of BPIV3 genotype C? Based on the ML phylogenetic tree in the responses, the statement “all overseas strains of BPIV3 genotype C clustered into another cluster (Fig. 2)” is less rigorous. Since there are only 61 available HN gene sequences of BPIV3 genotype C, it would be better to generate the phylogenetic tree figure showing all these sequences with the compact arrangement.

Response 2: Thanks for your professional suggestions. We have displayed the HN gene sequences of all BPIV3 genotype C strains on the ML phylogenetic tree and improved the description of the results to make them more rigorous.

Point 3: Results, lines 159-161, “Structural modeling of the HN protein shows that the mutation was located in the α-helix and the HN protein was predicted to contain one transmembrane region, which was located at 34–56 aa (Fig.3)”. Which domain/part is the mutation/α-helix located or near to? Why the authors mention the transmembrane region here?

Response 3: Thanks for your professional suggestions. As predicted by the Smart program, the structural domains of HN protein range from 34-571 aa, and our mutation site is located within the structural domains, but there is a paucity of studies on the functional analysis of the structural domain of BPIV3. So, we did not analyze it in more detail, and we sincerely hope to get your understanding. In addition, another reviewer suggested we add the predicted location of the transmembrane region of the HN protein. We considered that this reviewer wants to know whether the mutation site in this study is located in the transmembrane region, because it is known that the transmembrane region of the HN protein is very important for the role of the virus in host cell localization, etc.

Point 4: In this revised version of the manuscript, though the authors have found some unique genetic characteristics in sequences of Chinese BPIV3 genotype C strains, the analyses and evidences to support the conclusion that Chinese BPIV3 strains has a unique evolutionary trend is inadequate.

Response 4: The aim of this study was to investigate the prevalence and molecular characteristics of BPIV3 in most provinces in China. The results of this study showed that the sequences obtained by amplification were all BPIV3 genotype C strains, and we did our best to analyze these genotype C strains and found some unique genetic characteristics of genotype C strains. So now we changed the conclusion from "Chinese BPIV3 genotype C strains have unique evolutionary trends" to "Chinese BPIV3 genotype C strains have found some unique genetic characteristics."

Reviewer 3 Report

None

Author Response

Thank you for kindly reviewing the manuscript entitled “Prevalence and Molecular Characterization of Bovine Parainfluenza Virus Type 3 in cattle herds in China” (animals-2175528). We are grateful for the professional suggestions of the reviewers.